# Cytotoxic Properties of HT-2 Toxin in Human Chondrocytes: Could T_3_ Inhibit Toxicity of HT-2?

**DOI:** 10.3390/toxins11110667

**Published:** 2019-11-15

**Authors:** Feng’e Zhang, Mikko Juhani Lammi, Wanzhen Shao, Pan Zhang, Yanan Zhang, Haiyan Wei, Xiong Guo

**Affiliations:** 1School of Public Health, Health Science Center of Xi’an Jiaotong University, Key Laboratory of Trace Elements and Endemic Diseases, National Health Commission of the People’s Republic of China, Xi’an 710061, China; fenge0929@stu.xjtu.edu.cn (F.Z.); shaowz@stu.xjtu.edu.cn (W.S.); zhangpan891112@stu.xjtu.edu.cn (P.Z.); sahalasanmao@stu.xjtu.edu.cn (Y.Z.); ziyunpiaoxue700@stu.xjtu.edu.cn (H.W.); 2Department of Integrative Medical Biology, University of Umeå, 90187 Umeå, Sweden

**Keywords:** triiodothyronine, HT-2 toxin, cytotoxicity, Kashin-Beck disease

## Abstract

Thyroid hormone triiodothyronine (T_3_) plays an important role in coordinated endochondral ossification and hypertrophic differentiation of the growth plate, while aberrant thyroid hormone function appears to be related to skeletal malformations, osteoarthritis, and Kashin-Beck disease. The T-2 toxin, present extensively in cereal grains, and one of its main metabolites, HT-2 toxin, are hypothesized to be potential factors associated with hypertrophic chondrocyte-related osteochondropathy, known as the Kashin-Beck disease. In this study, we investigated the effects of T_3_ and HT-2 toxin on human chondrocytes. The immortalized human chondrocyte cell line, C-28/I2, was cultured in four different groups: controls, and cultures with T_3_, T_3_ plus HT-2 and HT-2 alone. Cytotoxicity was assessed using an MTT assay after 24-h-exposure. Quantitative RT-PCR was used to detect gene expression levels of *collagen types II* and *X*, *aggrecan* and *runx2*, and the differences in runx2 were confirmed with immunoblot analysis. T_3_ was only slightly cytotoxic, in contrast to the significant, dose-dependent cytotoxicity of HT-2 alone at concentrations ≥ 50 nM. T_3_, together with HT-2, significantly rescued the cytotoxic effect of HT-2. HT-2 induced significant increases in *aggrecan* and *runx2* gene expression, while the hypertrophic differentiation marker, *type X collagen*, remained unchanged. Thus, T_3_ protected against HT-2 induced cytotoxicity, and HT-2 was an inducer of the pre-hypertrophic state of the chondrocytes.

## 1. Introduction

Thyroid hormone (triiodothyronine, T_3_) is converted to this active form from thyroxine (T_4_) by deiodinase 2 (DIO2), and it is known to be an essential regulator in metabolism, growth, and development of the human body, and it is critical for the maturation of the skeletal system [1]. It controls the linear growth of bone by regulating endochondral ossification and promotes chondrocyte maturation and hypertrophic differentiation [2]. It has also been exploited to enhance cartilage formation and improve the functional properties of tissue-engineered neocartilage [3]. Furthermore, T_3_ enhances chondrogenesis of mesenchymal stem cells of the umbilical cord [4]. In addition, T_3_ regulates the transition between proliferation and terminal differentiation of chondrocytes in the growth plate via the *Wnt/β-catenin* signaling pathway [5,6]. Thus, a body of evidence implies that T_3_ has significant effects on cartilage and chondrocyte physiology. It is worth mentioning that up-regulated *DIO2* expression has been observed in osteoarthritic human articular cartilage and transgenic mice overexpressing *DIO2* [7]. Moreover, low serum T_3_ syndrome led to DIO2 dysfunction in Kashin-Beck disease (KBD) children [8,9].

T-2 and HT-2 toxins are two of the most representative and toxic members of the trichothecenes family, which are widely present in cereal grains and other cereal-based products, and are produced by various fungi species, such as Fusarium [10]. In rats, T-2 and HT-2 toxins were mainly distributed in the skeletal system at significantly higher concentrations than those in other organs [11]. In addition, the HT-2 toxin was shown to be a detectable metabolite of T-2 toxin in human chondrocytes, although it was deduced to be less toxic than T-2 [12]. After ingestion, the T-2 toxin is converted into more than 20 metabolites in animals [13]. The T-2 toxin is a cytotoxic fungal secondary metabolite produced by various species of Fusarium, and it interferes especially with the immune system, can harm fetal tissues, and induces cell death by apoptosis [13]. Furthermore, both the T-2 toxin and HT-2 toxin can result in apoptosis of chondrocytes by increased oxidative stress, which causes a release of *Bax*, *caspase-3*, and *caspase-9* [14]. A number of studies have reported that the T-2 toxin induces chondrocytes’ apoptosis, promotes catabolism and intracellular impairment of cartilage, and is a risk factor of KBD [15,16,17]. However, studies on the direct effects of the HT-2 toxin on cartilage and chondrocytes are still missing.

It is important to clarify the potentially damaging effect of the HT-2 toxin on human chondrocytes to enrich our knowledge of the possible molecular mechanisms of the HT-2 toxin causing cartilage lesions observed in KBD. Furthermore, this study aimed to explore whether T_3_ can protect from the chondrocytic injury caused by the HT-2 toxin in vitro, which may contribute to the combined effects both on cartilage and the potential pathogenesis of KBD. The concurrence of the abnormal T_3_ level and HT-2 toxin in vivo of KBD prompted this study to explore the effect of T_3_ and the HT-2 toxin on C-28/I2 chondrocytes and their combined effects.

## 2. Results

### 2.1. Individual Cytotoxicity of T_3_ and HT-2 Toxin in Human C-28/I2 Chondrocytes

MTT assay was used to evaluate the cytotoxicity in C-28/I2 chondrocyte cultures treated with T_3_ at concentrations ranging from 0 to 1000 nM. T_3_ was found to produce no major effect on the cell viability of C-28/I2 chondrocytes, even at 1000 nM concentration, although a statistically significant difference was observed at 50 nM (Figure 1A). In contrast, HT-2 was highly toxic to C-28/I2 cells, especially at concentrations ≥ 50 nM (Figure 1B).

### 2.2. T_3_ Protects against HT-2 Toxin-Induced Toxicity

Mixtures of HT-2 toxin and T_3_ at different concentration ratios of both were tested following 24-hour-long exposures. In general, HT-2 concentrations ≥50 nM significantly decreased the cell viability in comparison to control cultures (Figure 2). However, at equimolar concentrations, it took a 100 nM concentration of HT-2 to result in a significant decrease in the cell viability (Figure 2A). Also, when T_3_ was present at higher molar ratios in relation to HT-2, cytotoxicity was obvious at HT-2 toxin concentrations ≥50 nM (Figure 2B,C). At the ratio 1:1000, HT-2 concentration did not reach 50 nM concentration, and no decrease in cell viability was observed (Figure 2D).

When the molar concentration of HT-2 toxin was higher than T_3_, a decrease in the cell viability due to HT-2 toxin was obvious starting from 50 nM concentrations (Figure 2E–G). However, the addition of T_3_ could partially rescue the effect of HT-2 toxin on cell viability (Figure 2E). At high molar ration of HT-2 toxin, T_3_ did not have a protective effect on cell viability (Figure 2F,G). The dose-effect plot of all ratios generated by CompuSyn software is shown as the Appendix A.

### 2.3. Expressions of Extracellular Matrix and Hypertrophy Related-Genes in Chondrocyte Cultures Treated with T_3_ and/or HT-2 Toxin

The expression levels of four chondrocyte phenotype-related genes (*aggrecan*, *collagen types II* and *X*, and *runx2*) were quantified in C-28/I2 chondrocytes treated with 50 nM T_3_, 50 nM T_3_ plus 50 nM HT-2 toxin, or 50 nM HT-2 for 24 h. Compared with the control group, 50 nM T_3_ did not produce any significant changes in the gene expression levels of *collagen types II* and *X*, *aggrecan*, or *runx2* (Figure 3). The expression level of *aggrecan* was significantly increased in the presence of the HT-2 toxin (*p* < 0.05), and the level of *collagen type II* was 2-fold higher than the control (Figure 3). For the combination treatment of T_3_ and HT-2, gene expression patterns were similar to those by the HT-2 toxin alone (Figure 3).

It is well known that runx2 is a crucial transcription factor for chondrocyte maturation, and it induces the expression of *type X collagen* during the maturation process [18]. Thus, it was expected that T_3_, which takes part in hypertrophic differentiation, would increase the expression of *runx2* and *collagen type X*. However, neither were affected by T_3_ (Figure 3). Surprisingly, the HT-2 toxin significantly increased gene expression of *runx2*, although *type X collagen* expression remained unchanged from control levels. Immunoblot analysis was performed to confirm the induction of runx2 at the protein level. Indeed, immunoblotting confirmed the induced expression of runx2 produced by HT-2 (Figure 4). A slight, but not significant, increase in runx2 level was also observed for the T_3_ treatment. In conclusion, chondrocytes apparently reached the pre-hypertrophic stage in the presence of HT-2, while T_3_ could not promote this in the relatively short, 24-h-long time of the experiment. The indication that HT-2 would be such a strong inducer of runx2 expression was surprising.

## 3. Discussion

It is well known that T_3_ has an important role in growth plate maturation and development [1,2] and that inhibition of the T_3_ response by dominant-negative nuclear receptors promotes defects in cartilage maturation, ossification, and bone mineralization [19]. The regulation of T_3_ is particularly important during growth, which is the time when aberrations in endochondral ossification and growth occur in KBD [20]. Mycotoxins T-2 and its metabolite HT-2 have been shown to accumulate especially in the skeletal tissues [11], and they have been considered as possible factors for the KBD.

In this study, the cytotoxicities of T_3_ and the HT-2 toxin alone were first examined. At most, a very weak T_3_-mediated cytotoxicity in C-28/I2 chondrocytes was noticed, even at a non-physiologically high dose following 24-h of exposure. Also, a longer, 72-h-treatment, changed the response only minimally. In contrast, the HT-2 toxin had a cytotoxic effect on human chondrocytes at a 50 nM concentration after 24-h-exposure. In growth plate chondrocytes, long-term exposure to T_3_ inhibits cellular proliferation, which obviously is also related to hypertrophic differentiation [1].

When HT-2 toxin and T_3_ were administered at equal concentrations, the concentration to induce a statistically significant decrease in cell viability was shifted from 50 nM to 100 nM, indicating a protective effect of T_3_ on cytotoxicity induced by the HT-2 toxin. This led us to investigate how different molar ratios of the HT-2 toxin and T_3_ affect chondrocyte viability. When the ratio of HT-2:T_3_ was 10:1, HT-2 toxin concentrations ≥50 nM caused a significantly reduced cell viability, which was partly rescued by T_3_. At ratios 100:1 and 1000:1, there were no obvious combined effects on cell viability. 

Therefore, although T_3_ helped to reduce the cytotoxicity produced by the HT-2 toxin to human chondrocytes, it was most effective at an HT-2 toxin concentration range of 50–100 nM. However, the mechanism of the protective effect of T_3_ to chondrocyte death still remains unknown. In sheep growth plate chondrocytes, it has been shown that T_3_ is linked to chondrocyte proliferative capacity by targeted *FGFR3* to regulate telomerase reverse transcriptase expression and telomerase activity [21]. Also, the bone morphogenetic protein pathway has been implicated to be essential for the function of T_3_ in chondrogenesis [22].

To study the HT-2 and T_3_ effects on gene expression, we selected 50 nM T_3_ and 50 nM HT-2, since the 50 nM concentration of HT-2 was the lowest concentration that decreased the cell viability of cultured chondrocytes. It was also noticed that the HT-2 toxin induced an increase in gene expression of *aggrecan* and *runx2* and a trend for increased expression of *type II collagen*. The expression of *type X collagen*, a marker for hypertrophic chondrocytes [23], remained stable. Therefore, the cellular stage after HT-2 exposure can be considered to be pre-hypertrophic [18]. The increased level of runx2 at the protein level confirmed the mRNA result.

As mentioned previously, such a strong response in *aggrecan* and *runx2* by HT-2 in comparison to the T_3_ effect was surprising, since T_3_ is known to induce hypertrophic differentiation. In tissue engineering applications, T_3_ has been shown to increase the expression and synthesis of type II collagen [24], and improve articular cartilage surface architecture [25]. As the metabolite of T-2 toxin, it would be reasonable to assume that the HT-2 toxin will share similarity with the T-2 toxin, which leads to cartilage destruction by the degradation of the extracellular matrix [26,27]. In another study, the T-2 toxin promoted *aggrecanase-2* mRNA expression [28]. The *ROS-NFκB-HIF-2α* pathway was shown to be essential for the catabolic effects of the T-2 toxin [29]. However, in this cell culture model, it was not possible to confirm the possible anabolic or catabolic effects of HT-2 or T_3_ at the protein level due to the limited contents of extracellular matrix molecules secreted into the medium during the exposure time.

## 4. Conclusions

In conclusion, the HT-2 toxin led to significant cell death of human chondrocytes at rather low concentrations (threshold above 50 nM). Supplementation of T_3_ in cell culture medium decreased the cytotoxic effects of the HT-2 toxin only when it was applied in molar ratio 1:10, while other molar combinations failed to produce protective effects. The decrease in cell viability caused by the HT-2 toxin may be partly related to the finding that the cells appeared to shift quickly into a pre-hypertrophic state, indicated by an increased expression of *aggrecan* and *runx2*, and partly *collagen type II.* However, the major part of the HT-2 toxin effects is most likely due to its toxicity. Thus, further studies on the exact mechanism underlying the combined effect of T_3_ and HT-2 toxin on chondrocytes are warranted to provide a better understanding of the mechanism of HT-2 toxin cytotoxicity.

## 5. Materials and Methods 

### 5.1. Chondrocyte Culture

The immortalized human chondrocyte cell line C28/I2 was a kind gift from Dr. Mary B. Goldring (Hospital for Special Surgery, New York, NY, USA). Chondrocytes were cultured in Dulbecco’s modified Eagle’s medium (DMEM/F12; Hyclone, Logan, UT, USA), supplemented with 10% fetal bovine serum (FBS; Zhejiang Tianhang Biological Technology Stock Co., Huzhou, China), 100 U/mL penicillin/100 µg/mL streptomycin (Hyclone) at 37 °C and 5% CO_2_. During the culture period, the cells were passaged at subconfluency by sequential digestion in trypsin/EDTA (Hyclone; Sigma, St Louis, MO, USA), and the medium was replaced every 2 days [30]. Three to four independent experiments were performed.

### 5.2. MTT Cytotoxicity Assay

Human chondrocytes C-28/I2 were seeded in 96-well plates at a density of 6.5 × 10^3^ cells/well. After incubation for 24 h, the culture medium was replaced to fresh medium (DMEM/F12 with 10% FBS and 1% penicillin and streptomycin) containing several mixture concentrations of T_3_ and/or HT-2 toxin (Table 1), then treated for another 24 or 72 h. At the end of the intervention, the medium with T_3_ and/or HT-2 toxin was removed and replaced with fresh medium, and 20 μL aliquots of 5 mg/mL MTT stock solution (Amresco, Solon, OH, USA) were added into each well. After 4 h incubation in the presence of MTT to allow time for formazan formation, the medium was removed, and 150 μL dimethylsulfoxide was used to dissolve the formazan crystals from the wells. Optical densities of the samples were measured with a multi-plate reader (Infinite M200; Tecan Group, Männedorf, Switzerland) at a wavelength of 490 nm. The control groups included blank controls and normal controls. The blank control was fresh medium without the cells, and the normal control referred to the medium from the cell without T_3_ or the HT-2 toxin.

### 5.3. RNA Isolation and Quantitative Real-Time RT-PCR

Total RNA was isolated from the chondrocytes according to the manufacturer’s protocols using Trizol reagent (Invitrogen, Carlsbad, CA, USA). Reverse-transcription was performed using PrimeScript™ RT Master Mix Kit (Takara, Kusatsu, Shiga, Japan). Real-time PCR reactions were conducted in the Real-Time PCR Detection System (Bio-Rad, Hercules, CA, USA) using TB Green Premix Ex Taq™ II Kit (Takara), using the following parameters: 95 °C for 5 s, then 60 °C for 30 s, and 72 °C for 30 s, 40 cycles. The fold changes of relative gene expression were calculated with the 2^(−∆∆Ct)^ method [31] using *GAPDH* as the reference gene. The primer sequences used in this study are shown in Table 2.

### 5.4. Immunoblot Analysis

The total protein was extracted from the cultured chondrocytes according to the manufacturer’s protocols using an RIPA buffer (Beyotime, Shanghai, China). After denaturation, 30 μg of total protein was electrophoresed for immunoblot analysis. The blots were probed with primary antibodies directed against runx2 (Abcam, Cambridge, UK) overnight at 4 °C. GAPDH polyclonal antibody (Bioworld, Minneapolis, MN, USA) was used as a housekeeping reference. Peroxidase-conjugated goat anti-rabbit IgG (Thermo Fisher Scientific, Waltham, MA, USA) or goat anti-mouse IgG (Bioss, Shanghai, China) antibodies were used to visualize proteins using Western blotting chemiluminescence luminol reagent on a GeneGnome XRQ Western Blotting Analysis System (Syngene, Frederick, MD, USA). Working concentrations for each antibody were determined empirically based on the recommended stock solutions. Image J was used to quantify the band intensities of proteins of interest in the experimental and control groups.

### 5.5. Statistical Analysis

All experiments were performed three to four times. Parametric statistical analyses were selected to compare the effect of T3 and/or HT-2 toxin on C-28/I2 chondrocytes. One-way analysis of variance (ANOVA) was used to analyze the general difference, and the LSD-*t* (equal variances assumed) or Dunnett’s T3-test (equal variances not assumed) post-hoc tests were used for further pairwise comparison with SPSS 13.0 (SPSS Inc., Chicago, IL, USA). The difference was considered statistically significant when the *p*-value was less than 0.05.

## Figures and Tables

**Figure 1 toxins-11-00667-f001:**
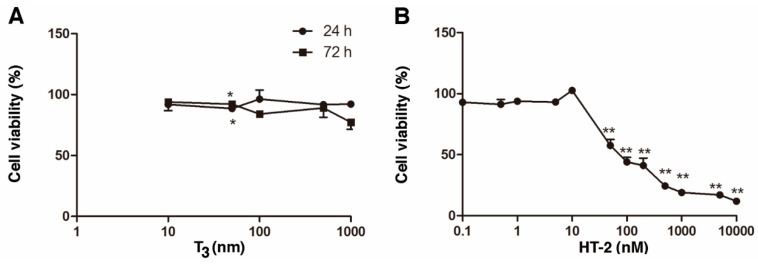
(**A**) Effect of T_3_ on viability of human C-28/I2 chondrocytes cultured for 24 and 72 h at T_3_ concentrations of 10, 50, 100, 500, and 1000 nM; (**B**) effect of HT-2 on human C-28/I2 chondrocytes cultured for 24 h at 0.1, 0.5, 1, 5, 10, 50, 100, 200, 500 1000, 5000, and 10,000 nM concentrations. The values show means ± SEM of three independent experiments. Cell viability of non-treated cultures in T_3_ experiments for 24 and 72 h were 100.0% ± 1.11% and 100.0% ± 10.06%, respectively, and in the HT-2 experiment, 100.0% ± 4.7%. Statistically significant differences against control cultures are indicated with asterisks, * *p* < 0.05 and ** *p* < 0.01.

**Figure 2 toxins-11-00667-f002:**
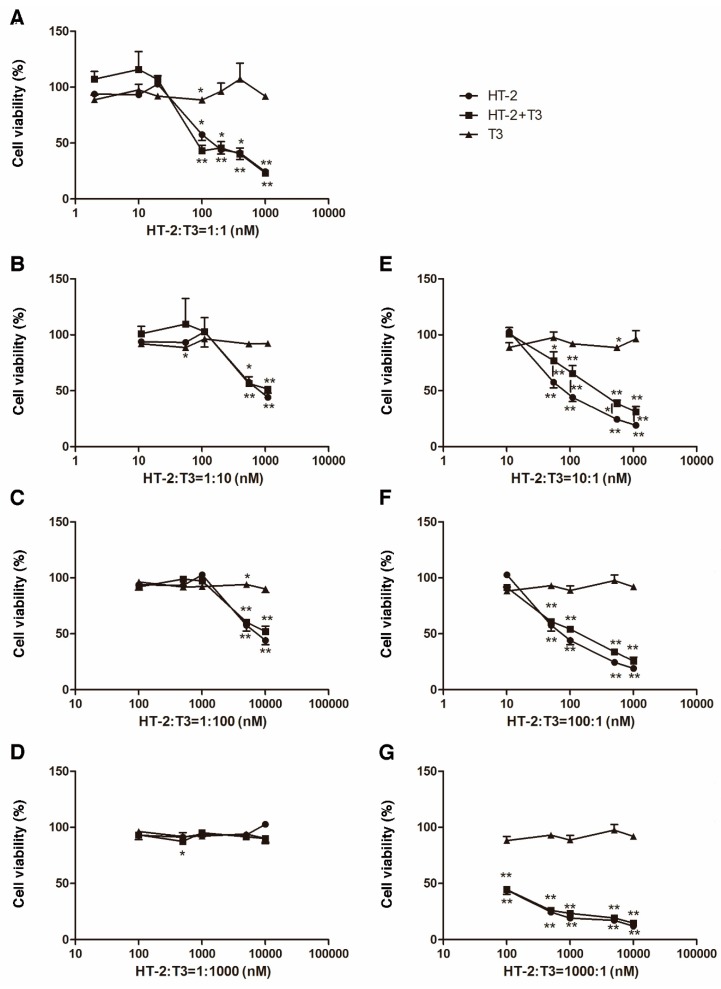
Cytotoxicity of T_3_ and the HT-2 toxin. The ratios of HT-2:T_3_ were (**A**) 1:1, (**B**) 1:10, (**C**) 1:100, (**D**) 1:1000, (**E**) 10:1, (**F**) 100:1, and (**G**) 1000:1. The values are shown as means ± SEM of three independent experiments. The cell viability in the non-treated control cultures were (**A**) 100.0% ± 4.5%, (**B**) 100.0% ± 3.7%, (**C**) 100.0% ± 5.0%, (**D**) 100.0% ± 4.1%, (**E**) 100.0% ± 0.4%, (**F**) 100.0% ± 1.3%, and (**G**) 100.0% ± 5.4%. Statistically significant differences against control cultures are marked with asterisks, * *p* < 0.05 and ** *p* < 0.01. The statistically significant differences observed between the HT-2 toxin and the mixture of the HT-2 toxin and T_3_ are also marked in (**E**–**G**). The amounts of T_3_ and HT-2 toxin for each mixture are provided in Table 1.

**Figure 3 toxins-11-00667-f003:**
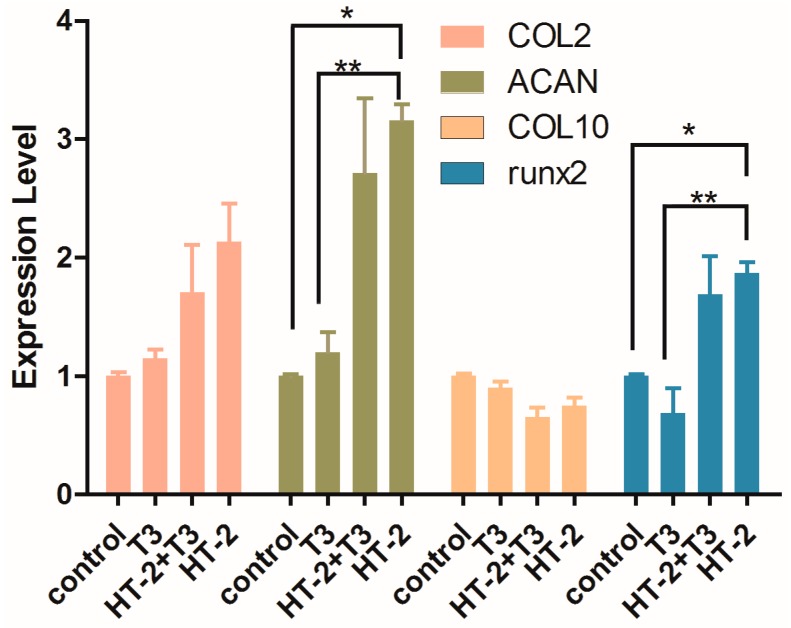
Gene expression levels of *collagen types II* and *X*, *aggrecan* and *runx2*, in C-28/I2 chondrocytes in control cultures and those treated with 50 nM T_3_ alone, 50 nM T_3_ plus 50 nM HT-2 toxin, and 50 nM HT-2 alone for 24 h. The fold changes are shown as mean ± SEM from three independent experiments. Statistically significant differences against control cultures are marked with asterisks, * *p* < 0.05 and ** *p* < 0.01.

**Figure 4 toxins-11-00667-f004:**
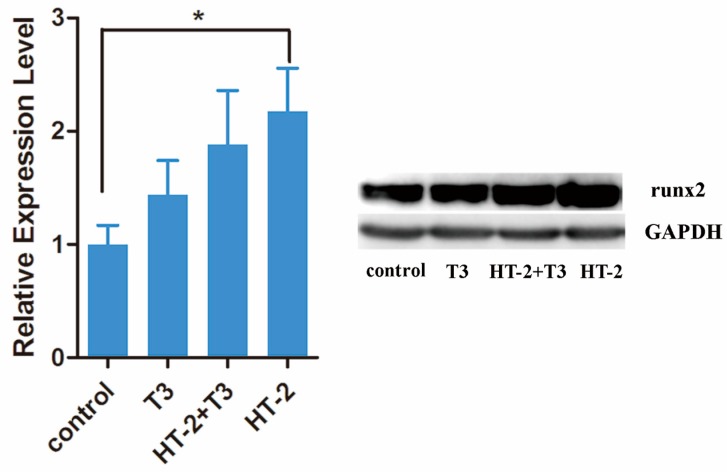
Protein expression levels of runx2 in C-28/I2 chondrocytes in control cultures and those treated with 50 nM T_3_ alone, 50 nM T_3_ plus 50 nM HT-2 toxin, and 50 nM HT-2 alone for 24 h. The fold changes are shown as mean ± SEM from four independent experiments. Statistically significant differences against control cultures are marked with asterisks, * *p* < 0.05.

**Table 1 toxins-11-00667-t001:** The contents of the HT-2 toxin and T_3_ mixtures.

Ratios	Components	Concentrations (nM)
**1:1**	HT-2	1, 5, 10, 50, 100, 200, 500
	T_3_	1, 5, 10, 50, 100, 200, 500
	HT-2:T_3_	1:1, 5:5, 10:10, 50:50, 100:100, 200:200, 500:500
**1:10**	HT-2	1, 5, 10, 50, 100
	T3	10, 50, 100, 500, 1000
	HT-2:T_3_	1:10, 5:50, 10:100, 50:500, 100:1000
**1:100**	HT-2	1, 5, 10, 50, 100
	T_3_	100, 500, 10, 50, 100, 200, 500
	HT-2:T_3_	1:100, 5:500, 10:1000, 50:5000, 100:10,000
**1:1000**	HT-2	0.1, 0.5, 1, 5, 10
	T_3_	100, 500, 1000, 5000, 10,000
	HT-2:T_3_	0.1:100, 0.5:500, 1:1000, 5:5000, 10:10,000
**10:1**	HT-2	10, 50, 100, 500, 1000
	T_3_	1, 5, 10, 50, 100
	HT-2:T_3_	10:1, 50:5, 100:10, 500:50, 1000:100
**100:1**	HT-2	10, 5,0 100, 500, 1000
	T_3_	0.1, 0.5, 1, 5, 10
	HT-2:T_3_	10:0.1, 50:0.5, 100:1, 500:5, 1000:10
**1000:1**	HT-2	100, 500, 1000, 5000, 10,000
	T_3_	0.1, 0.5, 1, 5, 10
	HT-2:T_3_	100:0.1, 500:0.5, 1000:1, 5000:5, 10,000:10

**Table 2 toxins-11-00667-t002:** Specific primers for quantitative RT-PCR.

Genes	Forward Primer	Reverse Primer
*COL2A1*	AGACTGGCGAGACTTGCGTCTA	ATCTCGGACGTTGGCAGTGTTG
*ACAN*	CTGAACGACAGGACCATCGAA	CGTGCCAGATCATCACCACA
*COL10A1*	GACTCATGTTTGGGTAGGCCTGTA	CCCTGAAGCCTGATCCAGGTA
*Runx2*	AGCTTCTGTCTGTGCCTTCTGG	GGAGTGGACGAGGCAAGAGTTT
*GAPDH*	GCACCGTCAAGGCTGAGAAC	TGGTGAAGACGCCAGTGGA

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
