# Peer review of "Cytotoxic Properties of HT-2 Toxin in Human Chondrocytes: Could T3 Inhibit Toxicity of HT-2?"

_toxins, 2019, doi:10.3390/toxins11110667_

Round 1
Reviewer 1 Report
The paper would contribute to our present knowledge on the effects of HT-2 toxins, the results are presented in a clear and concise manner.
Author Response
Dear reviewer,
We thank for the positive feedback on our manuscript submitted to Toxins. We have made some revisoins based on the comments given by other reviewers, and hope that they improve the manuscript in a approriate way.
Reviewer 2 Report
General comments: The manuscript represents results on cytotoxicity (MTT test) of hormone triiodothyronine (T3) and mycotoxin HT-2, alone and in combination in chondrocyte cell line, along with gene expression of collagen type II and X, aggrecan and runx2 (RT-PCR), involved in maturation of chondrocytes. In my opinion conclusions are only partially in line with results that have been obtained by experiments. Therefore, I have some remarks on methodology, statistics and conclusions that should be revised.
MAJOR REMARKS
Title is suggestive and does not reflect the results on cytotoxicity; combinations of various ratio HT-2 and T3 resulted in indifferent effect (based on the figure 2a,b,c,f,g), antagonising effect was observed only in combination HT-2:T3 (10:1) (figure 2e). Title should be changed into e.g. “Cytotoxic properties of triiodothyronine (T3) and HT-2 toxin in human chondrocytes: could T3 inhibit toxicity of HT-2?” or otherwise.
Materials and methods
MTT cytotoxicity assay: Are the cells during treatment with T3 and HT-2 were incubated in DMEM with or without serum? It is better to remove medium with toxin and T3 after treatment period and then add MTT reagent in medium because toxin could interact with MTT reagent and influence the absorbance reading. Is this 490 nm is correct wavelength? 490 is suitable for MTS test, for MTT is 570 or 540 nm. What was control in the experiment? Solvent of HT-2 and T3 should be controls?
In table 1 ratios of HT-2 and T3 are given, but there is no concentrations of the agents used in figure 1. These concentrations should be listed in materials and methods or below the figure 1.
Statistics: combinations should be presented as isobolograms which clearly shows the effect of combinations and based on that is possible to conclude on antagonising effect of HT-2:T3 (10:1) which suggest protective effect of T3 in particular combination.
Conclusions
Line 171-172 Supplementation of T3 in cell culture medium inhibited the cell death induced by HT-2 toxin under certain conditions should be changed into Supplementation of T3 in cell culture medium decreased cytotoxic effect of HT-2 toxin only when it was applied in molar ratio 1:10, while other molar combinations did not produced protective effect.
MINOR REMARKS
Lines 87, 140, 155 It si more appropriate to use the term cell viability instead of cell number
Author Response
General comments: The manuscript represents results on cytotoxicity (MTT test) of hormone triiodothyronine (T3) and mycotoxin HT-2, alone and in combination in chondrocyte cell line, along with gene expression of collagen type II and X, aggrecan and runx2 (RT-PCR), involved in maturation of chondrocytes. In my opinion conclusions are only partially in line with results that have been obtained by experiments. Therefore, I have some remarks on methodology, statistics and conclusions that should be revised.
MAJOR REMARKS
Title is suggestive and does not reflect the results on cytotoxicity; combinations of various ratio HT-2 and T3 resulted in indifferent effect (based on the figure 2a,b,c,f,g), antagonising effect was observed only in combination HT-2:T3 (10:1) (figure 2e). Title should be changed into e.g. “Cytotoxic properties of triiodothyronine (T3) and HT-2 toxin in human chondrocytes: could T3 inhibit toxicity of HT-2?” or otherwise.Response: We thank the reviewer for giving so useful suggestions. After careful discussion and consideration, we have changed the title into “Cytotoxic properties of HT-2 toxin in human chondrocytes: could T3 inhibit toxicity of HT-2?”, since T3 alone did not indeed have cytotoxic effects, in line 2-4, page 1.
Materials and methods
2.1 MTT cytotoxicity assay: Are the cells during treatment with T3 and HT-2 were incubated in DMEM with or without serum? It is better to remove medium with toxin and T3 after treatment period and then add MTT reagent in medium because toxin could interact with MTT reagent and influence the absorbance reading. Is this 490 nm is correct wavelength? 490 is suitable for MTS test, for MTT is 570 or 540 nm. What was control in the experiment? Solvent of HT-2 and T3 should be controls?
Response: Thanks for your proposal. The cells were incubated in DMEM with serum at 37°C and 5% CO2 during treatment with T3 and HT-2, we have now added the detail in line 206-207, page 6. We regret that the description of MTT was too short, we really have considered the influence of interaction between T3/HT-2 and MTT reagent, so at the end of the intervention, we added MTT reagent in fresh medium after removing medium with toxin and T3, the process was shown in line 208-209, page 6.
Acoording to our experience both 490 nm and 570 nm are suitable for MTT test. We use 490 nm as the detection wavelength for MTT, the previous studies about MTT can provide supportive evidence for this doubt, for example“Lin X, Shao W, Yu F, et al. Individual and combined toxicity of T-2 toxin and deoxynivalenol on human C-28/I2 and rat primary chondrocytes. Journal of Applied Toxicology, 2018”.
The controls included blank control and normal control. The blank control is tested in the well without cells which added only fresh medium, and the normal control refers to the cells cultured in fresh medium without either T3 or HT-2. We have added this description in line 214-216, page 7.
2.2 In table 1 ratios of HT-2 and T3 are given, but there is no concentrations of the agents used in figure 1. These concentrations should be listed in materials and methods or below the figure 1.
Response: Thanks for your advices. We have listed the concentrations of the agents used in figure 1 below the figure1, showed in lines 78-80, page 2.
2.3 Statistics: combinations should be presented as isobolograms which clearly shows the effect of combinations and based on that is possible to conclude on antagonising effect of HT-2:T3 (10:1) which suggest protective effect of T3 in particular combination.
Response: Thanks very much. We have tried to figure out the effect of combinations of HT-2 and T3 with the classic softwares CalcuSyn and CompuSyn. To our regret we cannot get the valid isobolograms and Fa-CI Plot for that both CalcuSyn and CompuSyn showed the results didn’t comply with the mathematical model. The dose-effect plot for figure 2 generated by CompuSyn software is shown in Attached file 1, and the results from CalcuSyn software are shown in Attached file 2. So we think that for the effect of combinations the data can be presented as the original Figure 2, which can refer to: “Individual and combined toxicity of T‐2 toxin and deoxynivalenol on human C-28/I2 and rat primary” by Lin X, Shao W, Yu F, et al.
3.Conclusions
3.1 Line 171-172 Supplementation of T3 in cell culture medium inhibited the cell death induced by HT-2 toxin under certain conditions should be changed into Supplementation of T3 in cell culture medium decreased cytotoxic effect of HT-2 toxin only when it was applied in molar ratio 1:10, while other molar combinations did not produced protective effect.
Response: Thank you for the professional advice, we have changed the sentence as you have given in line 185-187, page 6.
MINOR REMARKS
4.1 Lines 87, 140, 154 It is more appropriate to use the term cell viability instead of cell number.
Response: Thanks. We have changed the term cell number to cell viability, and also in Figure 1 and 2.
5. OTHER RESPONSES
The manuscript has also been read by native english-speaker, and text has been revised according to his suggestions.
Reviewer 3 Report
Comments to the Authors:
The manuscript entitled “Triiodothyronine protects against HT-2 toxin-induced chondrocyte cytotoxicity” have demonstrated significant research results of protecting activity of T3 towards chondrocytes in the presence of HT-2 toxin. Moreover, on the basis of available information, Authors try to partially explain the mechanism of this protection. The study was written carefully and well in terms of language.
Authors should correct manuscript according to the suggestion and completed some information.
Minor issues:
Line 42: in Introductions Authors should describe fungal producers of T-2 toxin
Line 47-48: Authors should described apoptosis mechanism of HT-2 towards chondrocytes
This publication should be helpful:
Yu, F-F., Lin, X-L., Wang, X., Ping, Z-G., Guo, X., Comparison of Apoptosis and Autophagy in Human Chondrocytes Induced by the T-2 and HT-2 Toxins, Toxins (basel), 2019 May; 11(5): 260, doi: 10.3390/toxins11050260
Line 47: “fusarium” it should be “ Fusarium”, pleas change in whole text
Line 132: in my opinion “factor” should be better than “candidates”
Line 189: it should be “6.5 x 103”
References:
Line 241, 255, 296 and 305: after title of journal comma should be deleted
Author Response
Comments to the Authors:
The manuscript entitled “Triiodothyronine protects against HT-2 toxin-induced chondrocyte cytotoxicity” have demonstrated significant research results of protecting activity of T3 towards chondrocytes in the presence of HT-2 toxin. Moreover, on the basis of available information, Authors try to partially explain the mechanism of this protection. The study was written carefully and well in terms of language.
Authors should correct manuscript according to the suggestion and completed some information.
1.Minor issues:
1.1. Line 42: in Introductions Authors should describe fungal producers of T-2 toxin
Response: Thank you for the reasonable advice. T-2 toxin is the representative of type B trichothecenes, produced on diverse cereal grains by various fungi species such as Fusarium. We have described fungal producers of T-2 toxin in line 49-50, page 2.
1.2 Line 47-48: Authors should described apoptosis mechanism of HT-2 towards chondrocytes
This publication should be helpful:
Yu, F-F., Lin, X-L., Wang, X., Ping, Z-G., Guo, X., Comparison of Apoptosis and Autophagy in Human Chondrocytes Induced by the T-2 and HT-2 Toxins, Toxins (basel), 2019 May; 11(5): 260, doi: 10.3390/toxins11050260
Response: We are grateful that you provide a very helpful rearch for us. After the summary, the induction of apoptosis by HT-2 toxin towards human chondrocytes involves Bax and caspase-3 signaling pathways. More concretely, both T-2 toxin and HT-2 toxin can result in apoptosis of chondrocytes by increased oxidative stress which causes the release of Bax, caspase-3, and caspase-9. We have described apoptosis mechanism of HT-2 towards chondrocytes in line 56-58, page.
1.3. Line 47: “fusarium” it should be “ Fusarium”, please change in whole text.
Response: Thank you for noticing this point. We have changed it in whole text.
1.4. Line 132: in my opinion “factor” should be better than “candidates”
Response: Thanks. We have changed the candidates to factor in line 144, page 5.
1.5 Line 189: it should be “6.5 x 103”
Response: We have changed “6500” to “6.5 x 103” in line 205, page 6.
2. References:
Line 241, 255, 296 and 305: after title of journal comma should be deleted
Response: Thank you for the important advice. We have deleted the comma after title of journal.
OTHER RESPONSES
The manuscript has also been read by native english-speaker, and text has been revised according to his suggestions.
Round 2
Reviewer 2 Report
Authors have included suggested changes into manuscript, in my opinion manuscript have been improved and it is suitable for publication in Toxins.